# Estimating the proportion of clinically suspected cholera cases that are true *Vibrio cholerae* infections: A systematic review and meta-analysis

**Kirsten E. Wiens**[1,2], **Hanmeng Xu**[1], **Kaiyue Zou**[1], **John Mwaba**[3,4,5], **Justin Lessler**[1,6,7], **Espoir Bwenge Malembaka**[1,8], **Maya N. Demby**[1], **Godfrey Bwire**[9], **Firdausi Qadri**[10], **Elizabeth C. Lee**[1], **Andrew S. Azman**[1,11,12]*

1 Department of Epidemiology, Johns Hopkins Bloomberg School of Public Health, Johns Hopkins University, Baltimore, Maryland, United States of America, 2 Department of Epidemiology and Biostatistics, College of Public Health, Temple University, Philadelphia, Pennsylvania, United States of America, 3 Centre for Infectious Disease Research in Zambia (CIDRZ), Lusaka, Zambia, 4 Department of Biomedical Sciences, School of Health Sciences, University of Zambia, Lusaka, Zambia, 5 Department of Pathology and Microbiology, University Teaching Hospital, Lusaka, Zambia, 6 Department of Epidemiology, Gillings School of Global Public Health, University of North Carolina at Chapel Hill, Chapel Hill, North Carolina, United States of America, 7 Carolina Population Center, University of North Carolina at Chapel Hill, Chapel Hill, North Carolina, United States of America, 8 Center for Tropical Diseases and Global Health (CTDGH), Université Catholique de Bukavu, Bukavu, Democratic Republic of the Congo, 9 Division of Public Health Emergency Preparedness and Response, Ministry of Health, Kampala, Uganda, 10 Infectious Diseases Division, International Centre for Diarrhoeal Disease Research Bangladesh (icddr,b), Dhaka, Bangladesh, 11 Geneva Centre for Emerging Viral Diseases, Geneva University Hospitals, Geneva, Switzerland, 12 Division of Tropical and Humanitarian Medicine, Geneva University Hospitals, Geneva, Switzerland

* azman@jhu.edu

**Data Availability Statement:** All input data and analytical code are available at https://github.com/HopkinsIDD/cholera_positivity.

**Funding:** This work was supported by the Bill and Melinda Gates Foundation (https://www.

## Abstract

### Background

Cholera surveillance relies on clinical diagnosis of acute watery diarrhea. Suspected cholera case definitions have high sensitivity but low specificity, challenging our ability to characterize cholera burden and epidemiology. Our objective was to estimate the proportion of clinically suspected cholera that are true *Vibrio cholerae* infections and identify factors that explain variation in positivity.

### Methods and findings

We conducted a systematic review of studies that tested ≥10 suspected cholera cases for *V. cholerae* O1/O139 using culture, PCR, and/or a rapid diagnostic test. We searched PubMed, Embase, Scopus, and Google Scholar for studies that sampled at least one suspected case between January 1, 2000 and April 19, 2023, to reflect contemporary patterns in *V. cholerae* positivity. We estimated diagnostic test sensitivity and specificity using a latent class meta-analysis. We estimated *V. cholerae* positivity using a random-effects meta-analysis, adjusting for test performance. We included 119 studies from 30 countries. *V. cholerae* positivity was lower in studies with representative sampling and in studies that set minimum ages in suspected case definitions. After adjusting for test performance, on

gatesfoundation.org/) [grant number OPP1171700 to A.S.A.] and the National Institute of Allergy and Infectious Disease (https://www.niaid.nih.gov/) [grant number AI135115-01A1 to A.S.A.]. The funders had no role in study design, data collection and analysis, decision to publish, or preparation of the manuscript.

**Competing interests:** JL is a paid statistical advisor for PLOS Medicine.

**Abbreviations:** CI, confidence interval; CrI, credible interval; GTFCC, Global Task Force on Cholera Control; IQR, interquartile range; JAGS, Just Another Gibbs Sampler; PCR, polymerase chain reaction; RDT, rapid diagnostic test.

average, 52% (95% credible interval (CrI): 24%, 80%) of suspected cases represented true *V. cholerae* infections. After adjusting for test performance and study methodology, the odds of a suspected case having a true infection were 5.71 (odds ratio 95% CrI: 1.53, 15.43) times higher when surveillance was initiated in response to an outbreak than in non-outbreak settings. Variation across studies was high, and a limitation of our approach was that we were unable to explain all the heterogeneity with study-level attributes, including diagnostic test used, setting, and case definitions.

## Conclusions

In this study, we found that burden estimates based on suspected cases alone may overestimate the incidence of medically attended cholera by 2-fold. However, accounting for cases missed by traditional clinical surveillance is key to unbiased cholera burden estimates. Given the substantial variability in positivity between settings, extrapolations from suspected to confirmed cases, which is necessary to estimate cholera incidence rates without exhaustive testing, should be based on local data.

## Author summary

### Why was this study done?

- Cholera surveillance typically relies on the clinical diagnosis of acute watery diarrhea (i.e., "suspected cholera"), but this definition has a low specificity for cholera.

- Our goal was to estimate the proportion of suspected cholera cases that are true *Vibrio cholerae* infections and identify factors that contribute to variation in observed positivity.

### What did the researchers do and find?

- We conducted a systematic review of studies from 2000 to 2023 that tested suspected cholera cases for *V. cholerae* infection using one of 3 different laboratory tests.

- We included 119 studies from 30 countries and found that, on average, half of suspected cholera cases represented true *V. cholerae* infections, after accounting for laboratory test accuracy.

- We also found high variability between studies and that the odds of a suspected case being a true infection were higher during outbreaks compared to non-outbreak settings.

### What do these findings mean?

- Our findings suggest that burden estimates based solely on suspected cases may overestimate the incidence of medically attended cholera by 2-fold.

- The high variability across studies suggests also that local testing data should be used to inform assumptions about positivity when exhaustive testing is not feasible.

- A limitation of our approach was that we could not account for cases missed by clinical surveillance, which is crucial for unbiased overall cholera burden estimates and an important area for future work.

## Introduction

Current estimates of cholera burden rely on clinical diagnosis of individuals with acute watery diarrhea (i.e., suspected cholera cases) [1,2]. It is unclear how many *Vibrio cholerae* O1/O139 (serogroups that cause current epidemics) infections get missed due to mild symptoms and other barriers to care-seeking or how many get overcounted due to nonspecific suspected case definitions. In Bangladesh, previous studies estimated that asymptomatic and unreported infections account for at least half of *V. cholerae* infections [3–5]. Meanwhile, the proportion of suspected cholera cases that represent laboratory-confirmed infections varies widely between studies, from 6% of those tested during routine surveillance in Bangladesh [6] to 72% of those tested during the initial phase of the 2017 outbreak in Yemen [7].

This wide variation in positivity may be caused by differences between sites in *V. cholerae* epidemiology [8], epidemiology of non-cholera diseases causing the same clinical symptoms [9–12], and variations in diagnostic tests and case definitions [13–15]. Typical suspected cholera case definitions have been shown to have high sensitivity but low specificity [14] for detecting true cholera and can vary by location across seasons [13]. Culture-based methods or polymerase chain reaction (PCR) are the gold standards to confirm cholera in clinical samples and generally have high specificity. Lateral flow rapid diagnostic tests (RDTs) may also be used and can be as sensitive as PCR [16]. Although recommended by the Global Task Force on Cholera Control (GTFCC) [17], systematic microbiological confirmation in surveillance is not always implemented, particularly during outbreaks when resources are limited [8]. To our knowledge—based on a literature review and discussion with experts—no study had yet systematically synthesized these data to estimate overall *V. cholerae* positivity and identify sources of this variation.

Understanding *V. cholerae* positivity among clinical cases could provide insights needed to improve laboratory testing strategies and allow for better estimates of cholera burden and risk, which are often used to allocate cholera resources, including oral cholera vaccines. Starting in 2023, the GTFCC has recommended using a combination of suspected cholera incidence, persistence, mortality, and cholera test positivity data across multiple years to identify priority areas for multisectoral interventions [18], which is particularly relevant in cholera endemic areas. As described above, the *V. cholerae* positivity data are often not available. We sought to address this knowledge gap by modeling the relationship between clinically suspected and laboratory confirmed cholera. Specifically, we aimed to estimate the proportion of suspected cholera cases that represent true *V. cholerae* O1/O139 infections and identify factors that explain variability in positivity across settings.

## Methods

### Ethics

This study was approved by the Johns Hopkins University Institutional Review Board and Temple University Institutional Review Board.

## Terminology

We focused on *V. cholerae* O1 and O139 because these are the serogroups that are responsible for the current seventh pandemic and the only ones known to lead to large outbreaks in humans [19]. These are also the serogroups that are targeted by each of the commonly used *V. cholerae* diagnostic tests (culture, PCR, and RDT). Throughout this manuscript, we refer to the proportion of suspected cholera cases that represent true *V. cholerae* O1/O139 infections as "*V. cholerae* positivity" or "cholera positivity." In addition, since the available data did not allow us to evaluate the performance of multiple RDTs, we refer to RDT as any rapid diagnostic test for *V. cholerae* O1/O139 and do not distinguish between different RDT manufacturers or whether the RDT is enriched/direct swab RDT or stool RDT.

## Systematic review

This study is reported as per the Preferred Reporting Items for Systematic Reviews and Meta-Analyses (PRISMA) guideline (S1 Checklist). The review was not preregistered, and a formal public protocol was not prepared, although all study methods can be found in the Methods below and the Methods in S1 Appendix.

We searched PubMed, Embase, Scopus, Google Scholar, and *medRxiv* on October 16, 2021, using search provided in the Supplementary Methods in S1 Appendix. We updated PubMed, Embase, and Scopus searches on April 19, 2023. We included studies that (1) collected human samples; (2) reported the number of suspected and confirmed cholera cases in the sampling frame; (3) used culture, PCR, and/or RDT to test suspected cases for cholera; and (4) had at least one suspected case sample collected on or after January 1, 2000, to reflect contemporary patterns in cholera positivity. We excluded studies that (1) used a case definition not specific for suspected cholera (i.e., we accepted non-bloody watery diarrhea, acute watery diarrhea, or simply suspected cholera but not diarrhea, acute diarrhea, or acute gastroenteritis); (2) sampled only special populations (i.e., people living with HIV or cancer); (3) selected suspected cases based on epidemiological link to other cases or environmental sources; (4) tested fewer than 10 suspected cases; and (5) were reported in languages other than English, French, Spanish, and Chinese (languages our study team had proficiency in). We did not exclude studies based on study type or sampling method. Although we originally included preprints in our screening and extracted one preprint, we excluded this study at the time of the updated search because the published version of the manuscript no longer included positivity data.

Titles, abstracts, and full texts were uploaded to Covidence, a web-based screening tool (https://www.covidence.org/), and were assessed independently by two of the reviewers (ASA, ECL, HX, KEW, KZ, MND) for inclusion. Conflicts were resolved either by a third reviewer or through consensus/discussion. Data were extracted from included studies in a shared spreadsheet (S1 Data) by a single reviewer. The key extracted items included study timeframe and location, surveillance type (routine, outbreak, post-vaccination, or hybrid), case definition of suspected cholera (including age constraint and whether dehydrated or hospitalized, if provided), test method(s), sampling strategy for the test (all suspected cases, systematic or random sampling, convenience sampling, or unreported), number of tested and confirmed suspected cases, among other sample characteristics, if included. If only the proportion positive and total number tested were reported, the number of confirmed cholera cases was calculated by hand and rounded to the nearest whole number. If the surveillance contained multiple timeframes, tested samples with multiple tests, or reported stratified results, we extracted the data separately into different rows in the spreadsheet.

To identify overlapping samples, we manually reviewed all studies with overlapping timeframes by country. We excluded studies that had shorter timeframes, fewer suspected cases

tested, less representative sampling methods, fewer confirmation tests, or reported positive results by 2 tests but did not disaggregate. Within studies, when suspected and confirmed cases were stratified multiple ways, we included the stratification by surveillance type if available, followed by age, antibiotic use, dehydration status, year, geography, or sex, in that order. When studies used multiple RDTs, we included results for Crystal VC (Arkray Healthcare, Gujarat, India) and direct rapid tests (as opposed to rapid tests performed after an enrichment step) because these were the most common.

To identify any mistakes and ensure quality of the extracted data, we performed data quality checks using a series of automated functions in R to identify implausible values (e.g., start date of study after end date, more cases positive than tested, lower age limit larger than upper age limit) and missing required data. If impossible or missing values were found, the entire extraction was double checked for accuracy and corrected by a single reviewer.

To assess whether different studies used methodologies that may have biased our results, we plotted cholera positivity in the raw data by (1) diagnostic test used; (2) sampling method quality; and (3) suspected cholera case definition. In addition, we plotted the relationship between cholera positivity in the raw data and (1) estimated suspected cholera incidence [2]; (2) the proportion of cases severely dehydrated; and (3) the proportion on antibiotics. We quantified the correlation between these variables using Spearman's rank correlation coefficient using the spearman.ci function of the RVAideMemoire package in R [20]. Since these continuous variables were only available in a subset of studies, we did not adjust for them in final analyses. All data visualization was conducted using the ggplot2 package in R [21].

## Data analysis

**Estimating sensitivity and specificity of cholera confirmation tests.** We constructed a latent-class model to assess sensitivity and specificity of culture, PCR, and RDT, assuming none had perfect performance. We fit a hierarchical conditional dependence model, similar to that proposed by Wang and colleagues, which takes into account potential pairwise dependence between the tests that could occur if the tests have reduced performance for similar reasons [22]. We performed inference in a Bayesian framework using Just Another Gibbs Sampler (JAGS) through the rjags package in R [23,24]. We pooled estimates across 4 published studies that reported cholera confirmation results for all 3 test methods [16,25–27].

We used flat prior distributions on sensitivity and specificity of each test with a lower bound set based on plausible values from the literature [15,16,25–27] (Table A in S1 Appendix). We assumed that culture had lower sensitivity than PCR and RDT because it depends on successful growth of viable *V. cholerae* in the laboratory. We assumed that RDT had lower specificity than culture and PCR because it may have cross-reactivity with other antigens in the stool or defects that lead to false positive results. For each prior, we selected a wider range than had been reported in previous studies to allow for greater variation. We ran 4 chains of 100,000 iterations and assessed convergence through visual inspection of traceplots and with the Gelman-Rubin R-hat statistic.

**Estimating *V. cholerae* positivity and sources of heterogeneity.** We pooled estimates of *V. cholerae* positivity across all studies using a generalized linear model with a study level random intercept, which allowed us to adjust for sensitivity and specificity of the diagnostic tests as well as examine the contributions of study methodology (i.e., whether the study used low- versus high-quality sampling, and whether or not the study set a minimum age in the suspected cholera case definition) and setting (whether surveillance was routine or post-vaccination versus initiated in response to an outbreak) on variation in positivity. To estimate the proportion positive, overall and by strata, we marginalized over study-level random effects.

See Supplementary Methods in S1 Appendix for the full statistical model. We performed inference in a Bayesian framework using CmdStanR version 0.5.2 as an interface to Stan for R [24,28]. We additionally performed a sensitivity analysis where we shifted the prior set on the global intercept (see Methods in S1 Appendix). The odds of a suspected cholera case having a true *V. cholerae* infection given each covariate were calculated as odds ratios by taking the mean and 95% credible interval (CrI) of 8,000 draws from the posterior distribution of each covariate's exponentiated coefficient. Odds ratios with 95% CrIs that did not cross the value 1 were considered statistically significant.

To estimate the proportion of the variance in positivity attributable to true differences between studies, beyond simple sampling error, we calculated the $I^2$ statistic [29] as

$$I^2 = \frac{\tau^2}{\tau^2 + \upsilon}$$

where $\tau^2$ was between-study heterogeneity or the variance of the random effect by observation. We calculated the within-study variance, $\upsilon$, [30] as

$$\upsilon = \frac{(k-1)\sum_1^i \omega_i}{\left(\sum_1^i \omega_i\right)^2 - \sum_1^i \omega_i^2}$$

where $k$ was the number of studies or observations included in the meta-analysis, and $\omega_i = 1/\nu_i$ where $\nu_i$ was the variance of the proportion positive by culture, PCR, or RDT within each study/observation. When multiple tests were used in a study, we used the maximum variance estimate across the tests.

## Results

### Study characteristics

We identified 131 studies that met our inclusion criteria (Fig 1). Of these, 119 studies contained nonoverlapping samples and were included in our analysis dataset [6,7,9,10,12–14,16,25–27,31–131] and 12 were excluded from analysis due to overlaps [8,11,132–141] (Fig 1). Of the 119 studies included in our analysis dataset, one reported data for more than one sampling method [7], one for both outbreak and non-outbreak surveillance [37], and one for outbreak and non-outbreak surveillance in 6 different countries [13]. We defined each of these as separate entries in the dataset for a total of 132 observations. Extracted data including detailed individual study information can be found in S1 Data.

The nonoverlapping observations in our analysis dataset came from 30 countries and were reported at different geographic levels, including the country level ($n = 16$ observations) and first ($n = 25$), second ($n = 66$), and third administrative levels ($n = 25$) (Fig A in S1 Appendix). Twelve studies reported data for multiple administrative units, and 3 reported across multiple administrative divisions within a country; the numbers above reflect the largest administrative division reported per observation. Data were collected from 1992 through 2022 with most observations from studies that completed sampling during 2015 to 2022 ($n = 53$ observations), followed by 2010 to 2014 ($n = 32$), 2005 to 2009 ($n = 21$), and 1997 to 2004 ($n = 17$) (Fig B in S1 Appendix). Nine studies were missing sampling end dates. Most studies were conducted in South Asia and West, Central, and East Africa, with additional studies from Haiti, Yemen, Iraq, Iran, Laos, Vietnam, Papua New Guinea, Algeria, and the Philippines (Fig A in S1 Appendix).

Most of the observations were from surveillance studies (93/132, 70.5%), followed by diagnostic test accuracy studies (28/132, 21.2%) and vaccine effectiveness studies (10/132, 7.6%)

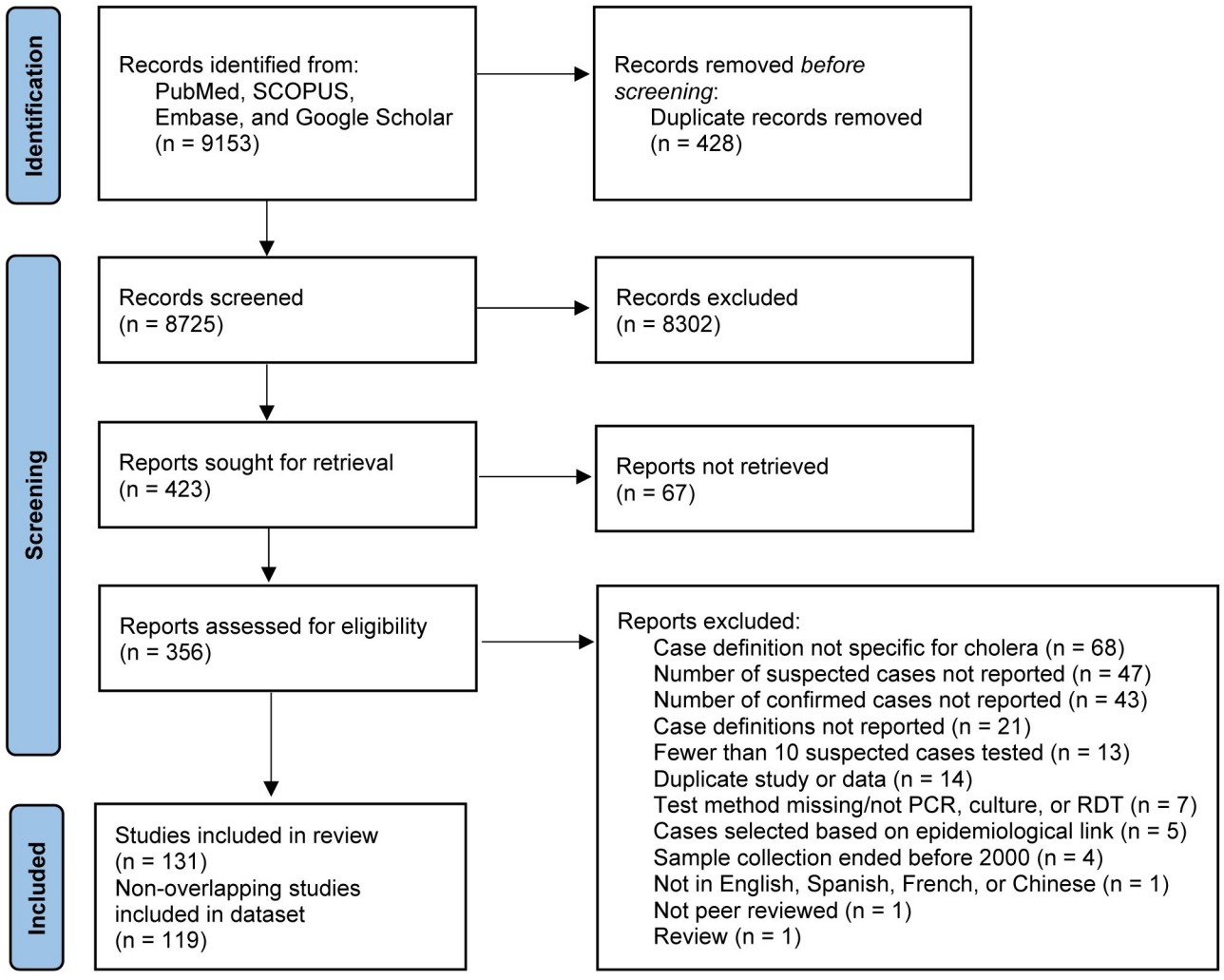

**Fig 1. PRISMA flow diagram.** Diagram illustrating literature selection process, including databases searched, literature screened, and full texts reviewed for eligibility. Reasons for exclusion are indicated along with the number of studies that fell within each category. PCR, polymerase chain reaction; RDT, rapid diagnostic test.

(Table 1). High-quality sampling methods (i.e., tested all suspected cases, a random sample, or systematically selected every nth suspected case) were used in 28% (37/132) of observations, while the remaining 72% (95/132) used convenience sampling or did not report the sampling approach (Table 1). Even though most studies did not include *V. cholerae* positivity disaggregated by individual-level characteristics, 24.2% (32/132) reported the proportion of suspected cases under age 5, 8.3% (11/132) reported the proportion severely dehydrated, 7.6% (10/132) reported the proportion on antibiotics, and one study reported all 3 (Table B in S1 Appendix).

## *V. cholerae* positivity in unadjusted data

We found that reported *V. cholerae* positivity varied greatly across studies with an interquartile range (IQR) of 30% to 60% ($N = 165$ observations of positivity; 25 of the 131 observations had positivity results for multiple tests) (Table 1). As expected, positivity varied by diagnostic test used with a median positivity of 36% by culture (IQR, 27% to 55%; $N = 121$), 37% by PCR (IQR, 34% to 55%; $N = 11$), and 49% by RDT (IQR, 38% to 67%; $N = 33$), with substantial

**Table 1. Study characteristics.** Number of observations included in the analysis dataset with each study characteristic. There is more than one observation per study when the study reported data for more than one sampling method, surveillance type, and/or country.

| Category | Characteristic | Number (*n* = 132) | Percent |
|---|---|---|---|
| Study design | Surveillance | 93 | 70.5 |
| | Diagnostic test accuracy | 28 | 21.2 |
| | Vaccine effectiveness | 10 | 7.6 |
| | Randomized control trial | 1 | 0.8 |
| Sampling method quality | High | 37 | 28.0 |
| | Low | 95 | 72.0 |
| Percent of suspected cases tested | 0–4 | 12 | 9.1 |
| | 5–49 | 32 | 24.2 |
| | 50–95 | 27 | 20.5 |
| | ≥95 | 30 | 22.7 |
| | Not reported | 31 | 23.5 |
| Number of tests used (of culture, PCR, and/or RDT)* | 1 | 106 | 80.3 |
| | 2 | 19 | 14.4 |
| | ≥3 | 7 | 5.3 |
| Number of suspected cases tested | 1–9[†] | 1 | 0.8 |
| | 10–99 | 37 | 28.0 |
| | 100–999 | 55 | 41.7 |
| | ≥1,000 | 39 | 29.5 |

*PCR, polymerase chain reaction; RDT, rapid diagnostic test.

[†]One multicountry surveillance study overall tested ≥10 suspected cholera cases for *V. cholerae* O1/O139 but reported fewer than 10 tested in one country.

overlap between distributions (Fig 2A). Positivity was higher across studies that used low-quality or convenience sampling methods (median of 43%; *N* = 117; IQR, 33% to 62%) compared to those that used high-quality or representative sampling (median of 35%; IQR, 14% to 51%) (Fig 2B). Positivity increased with higher minimum ages in suspected cholera case definitions (Fig 2C), and we found a modest negative correlation between positivity and the proportion of suspected cases under 5 years old (Spearman *r* = −0.60; 95% confidence interval (CI): −0.81, −0.32; *p* < 0.001) (Fig Ca in S1 Appendix).

Unadjusted positivity was higher when surveillance was initiated in response to an outbreak (median of 47%; IQR, 33% to 66%; *N* = 80) compared to situations where surveillance was routine or post-vaccination (median of 35%; IQR 17% to 49%; *N* = 85) (Fig 2D). We found limited evidence for differences in positivity by the 2010 to 2016 estimated mean annual suspected case incidence rate in countries where these estimates were available (Fig Cb in S1 Appendix; [2]).

We found a modest positive correlation between positivity and the proportion of suspected cases severely dehydrated (Spearman *r* = 0.64; 95% CI: 0.22, 0.90; *p* = 0.001) (Fig Cc in S1 Appendix). While not statistically significant, we found a weak negative correlation between positivity and the proportion of suspected cases that had received antibiotics prior to testing (Spearman *r* = −0.46; 95% CI: −0.83, 0.09; *p* = 0.07) (Fig Cd in S1 Appendix).

## Adjusted underlying *V. cholerae* positivity

Since different imperfect diagnostic tests were used to confirm *V. cholerae* O1/O139, we adjusted positivity estimates from each study to account for test performance. To estimate a median performance of each type of diagnostic test, we pooled estimates of sensitivity and specificity across 4 studies that reported detailed results for all 3 tests (see Methods). This included data from Bangladesh [27], South Sudan [16], Kenya [25], and Zambia [26]. We

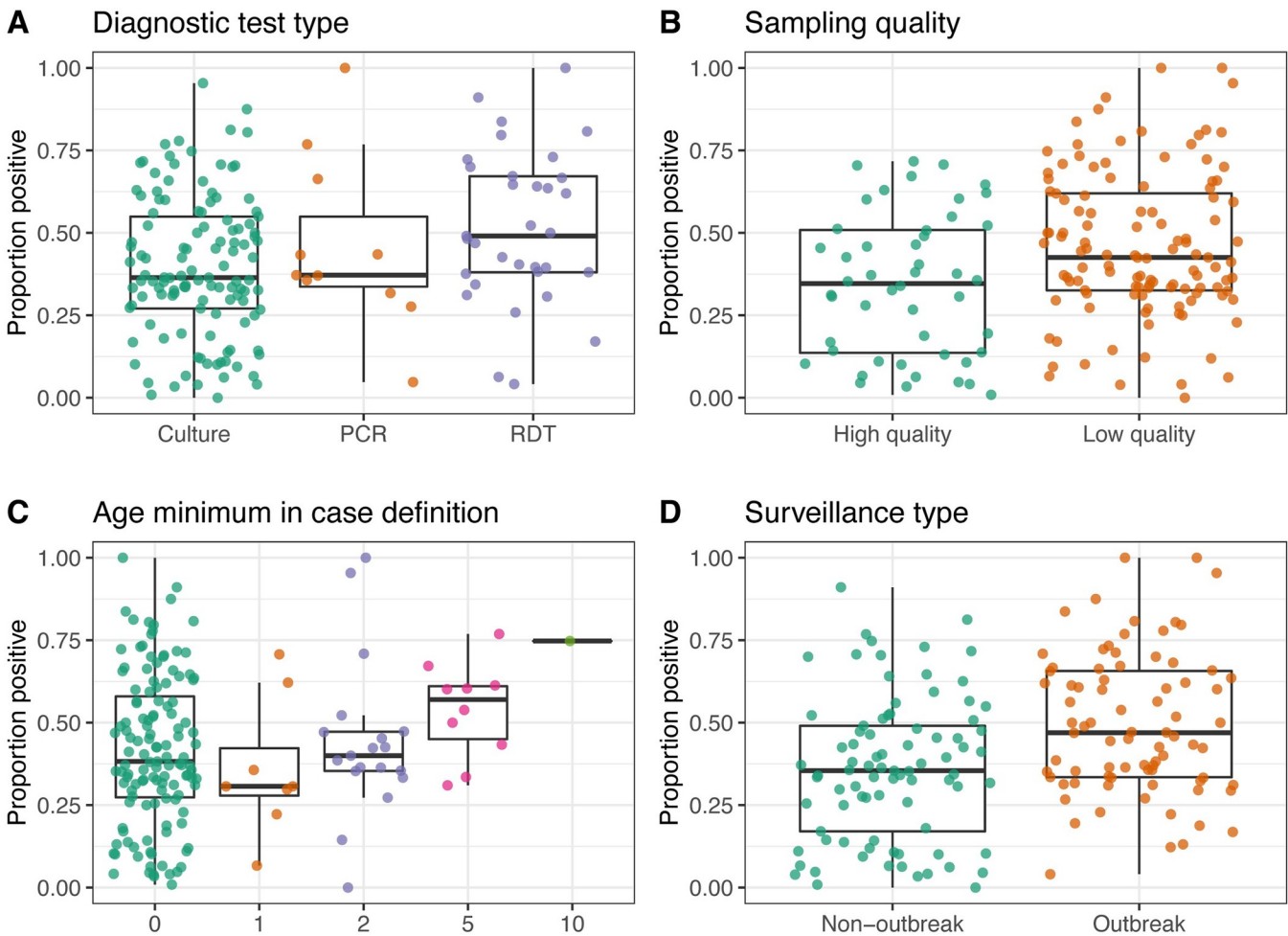

**Fig 2. *V. cholerae* positivity by study methodology and outbreak context.** Proportion of suspected cholera cases that were confirmed positive by **(A)** diagnostic test type, **(B)** quality of sampling methods, where "high" includes all suspected cases or a random or stratified sample and "low" includes convenience or unreported sampling methods, **(C)** age minimum in suspected case definition, where "0" indicates that no minimum age was set, and **(D)** whether surveillance was initiated in response to an outbreak or whether it was routine surveillance or non-outbreak. Each point is an observation included in the analysis dataset. There is more than one observation per study when the study reported data for more than one sampling method, surveillance type, and/or country. Boxes represent the median and IQR of positivity for each group. Lines extend from the top and bottom of box to the largest positivity value no further than 1.5 * IQR from the box. IQR, interquartile range; PCR, polymerase chain reaction; RDT, rapid diagnostic test.

estimated a median sensitivity of 82.0% (95% CrI: 37.5, 98.7) and specificity of 94.3% (95% CrI: 81.5, 99.6) for culture, a median sensitivity of 85.1% (95% CrI: 53.6%, 98.9%) and specificity of 94.2 (95% CrI: 81.8, 99.7) for PCR, and a median sensitivity of 90.4% (95% CrI: 55.2, 99.5) and specificity of 88.9% (95% CrI: 54.9, 99.4) for RDT (Fig 3A and Table C in S1 Appendix).

After adjusting for diagnostic test performance, we estimated that 53% (95% CrI: 24%, 80%) of suspected cases tested were true *V. cholerae* O1/O139 infections across all studies (Fig 3 and Fig D in S1 Appendix and Table D in S1 Appendix). These estimates remained similar in sensitivity analysis with an alternative prior distribution (Table D in S1 Appendix).

With additional adjustments for study methodology (i.e., sampling quality and whether an age minimum was set in suspected case definition), we estimated that *V. cholerae* positivity for studies with high-quality sampling methods was 46% (95% CrI: 19%, 76%) when no age restriction was used and 68% (95% CrI: 33%, 98%) when a minimum age (typically 1 or 5 years old) was incorporated into the case definition (Fig 3, Table D in S1 Appendix). After

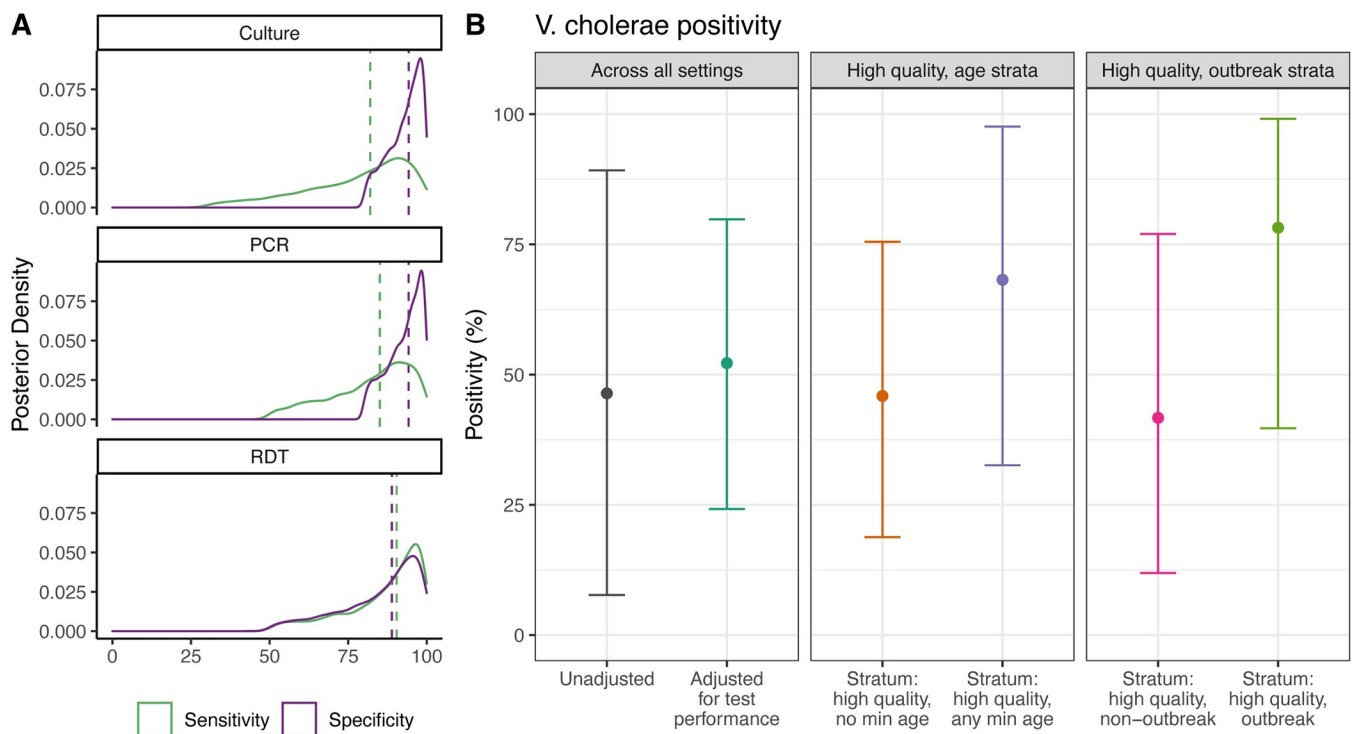

**Fig 3. Estimated underlying *V. cholerae* positivity. (A)** Posterior distributions of pooled percent sensitivity and specificity of culture (top), PCR (middle), and RDT (bottom) for detecting *V. cholerae* O1/O139 infections in suspected cholera cases. Dashed lines represent median values of each distribution. **(B)** The "Unadjusted" dot is mean *V. cholerae* positivity (lines represent 95% CrI) from random effects meta-analysis without adjustments for test performance. The "Adjusted for test performance" and "Stratum: . . ." dots are estimated mean *V. cholerae* positivity (lines represent 95% CrIs), adjusted for sensitivity/specificity of the tests. High-quality stratified estimates correspond to post-stratified estimates of *V. cholerae* positivity for studies that use high quality sampling methods and whether an age minimum was set in the suspected case definition, as well as whether surveillance was initiated in response to an outbreak. CrI, credible interval; PCR, polymerase chain reaction; RDT, rapid diagnostic test.

adjusting for sampling quality and whether or not surveillance was initiated in response to a cholera outbreak, we estimated that *V. cholerae* positivity for studies with high-quality sampling methods was 42% (95% CrI: 12%, 77%) in non-outbreak settings and 78% (95% CrI: 40%, 99%) in outbreak settings (Fig 3, Table D in S1 Appendix).

We found substantial heterogeneity between studies ($I^2$ = >99.99% (95% CrI: >99.99%, >99.99%; $I^2$ = 0.96 (95% CrI: 0.94, 0.98)) (Fig 4). Adjusted underlying positivity rates ranged from 0.008% (95% CrI: 0.0004%, 0.04%) for a high-quality study conducted during routine surveillance in Bangladesh to 99.8% (95% CrI: 98.7%, 100.0%) for a "low-quality" study conducted during a cholera outbreak in Uganda (Fig 4).

## Factors associated with variation in *V. cholerae* positivity

We then examined factors that could explain variation in *V. cholerae* positivity. After adjusting for test performance, sampling quality, and outbreak setting, we found that setting any minimum age in the case definition (i.e., 1, 2, 5, or 10) was associated with 2.33 (95% CrI: 0.54, 6.40) times higher odds of a suspected cholera case having a true infection (Table E in S1 Appendix).

We estimated that the odds of a suspected cholera case having a true *V. cholerae* O1/O139 infection were 5.71 (95% CrI: 1.53, 15.43) times higher when surveillance was initiated in response to a cholera outbreak compared to non-outbreak surveillance, after adjusting for test performance, sampling quality, and case definition (Table E in S1 Appendix).

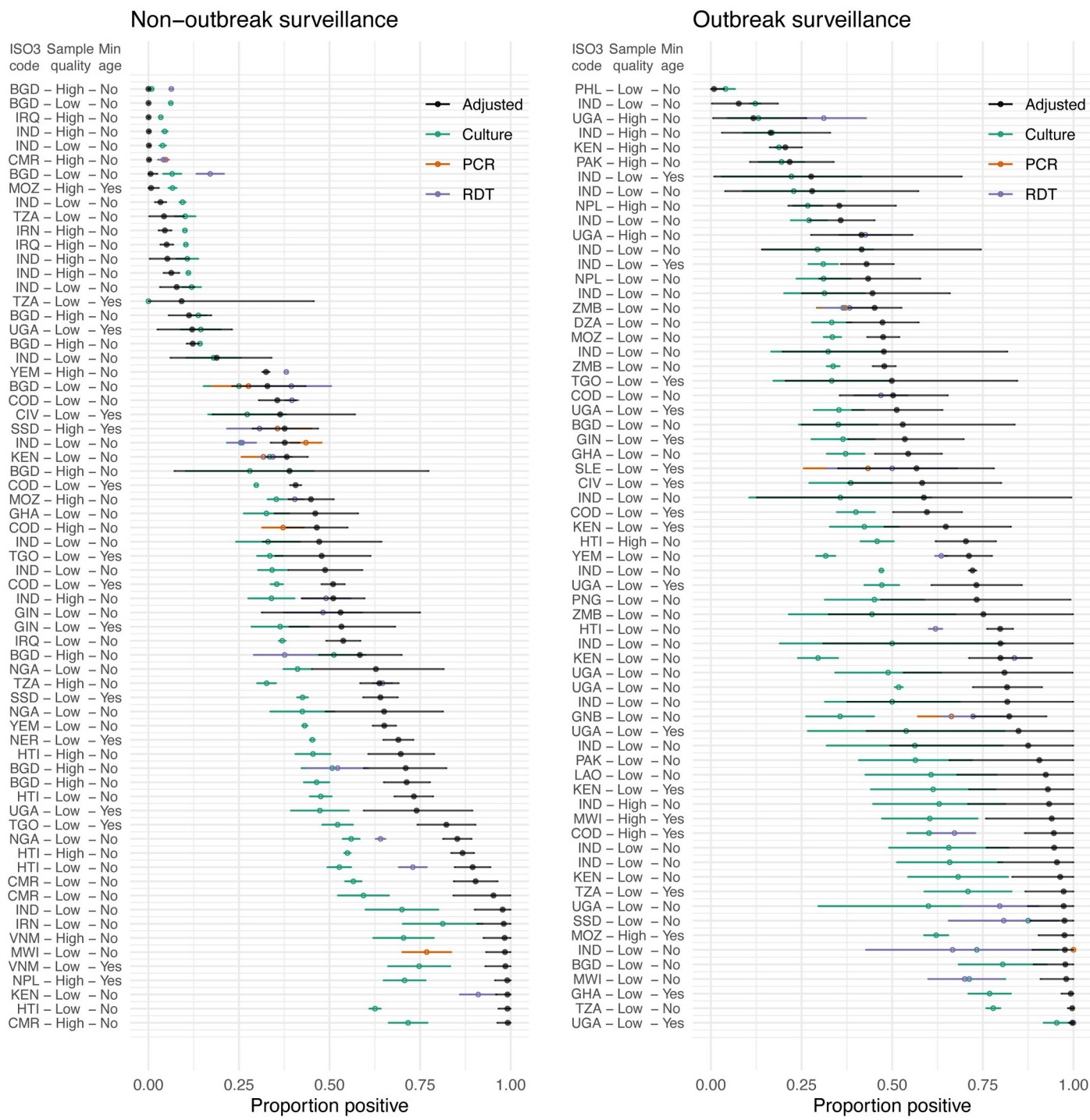

**Fig 4. Forest plot of study estimates and underlying positivity.** Black points indicate mean study-level underlying positivity and 95% CrI. Teal, orange, and purple points indicate the proportion positive reported by study for culture, PCR, and RDT, respectively, and corresponding error bars indicate 95% CI for a binomial probability using the normal approximation [147]. Studies are labeled by country ISO3 code, quality of sampling methods, (high or low), and whether a minimum age was set in the suspected cholera case definition, (yes or no). Studies are split into outbreak and non-outbreak for ease of interpretation. CI, confidence interval; CrI, credible interval; PCR, polymerase chain reaction; RDT, rapid diagnostic test.

## Discussion

Here, we estimated that, on average, half of medically attended suspected cholera cases represent true *V. cholerae* O1/O139 infections. We found that *V. cholerae* positivity was higher when a minimum age was set in case definitions and when surveillance was initiated in response to an outbreak. Additionally, we found substantial heterogeneity in *V. cholerae* positivity between studies, so that simply multiplying the number of suspected cholera case counts by this global proportion positive to estimate the true number of cases will not be appropriate in most settings. To our knowledge, this is the first study to systematically synthesize data globally to estimate overall *V. cholerae* positivity and examine factors that contribute to variation in positivity.

A remaining question is why only about half of medically attended suspected cholera cases represent true infections. It is possible that we overestimated test sensitivity and have not fully accounted for false negatives; unfortunately, this is difficult to evaluate without a gold standard diagnostic test. A portion of the remaining suspected cases could also be infections with other enteric pathogens, especially those with similar transmission modes as cholera that may have outbreaks or high levels of endemic transmission concurrently. For example, in Uvira, Democratic Republic of the Congo, 36% of suspected cholera cases were positive for Enterotoxigenic *Escherichia coli* and 28% for *Cryptosporidium* [10]. In rural Bangladesh, the majority of acute watery diarrhea in children under 18 months was attributable to rotavirus, while older children were more often infected with *V. cholerae* [12]. In Haiti, 64% of acute watery diarrhea cases tested positive for *V. cholerae* O1, 4% for rotavirus, and <1% for Shigella and Salmonella, though rotavirus positivity was higher among children under 5 [11]. Thus, the relative contribution of non-cholera watery diarrhea varies with age distribution and other location-specific drivers of enteric infections.

One of the limitations of this study was that we could not account for all potential drivers of *V. cholerae* positivity, which contributed to the large heterogeneity we found between studies. In addition, *V. cholerae* positivity may be highest in the early stages of an outbreak [7,9,131], but we could not account for this, given the temporal resolution of our dataset. However, a strength of our approach is that we pooled estimates from studies across diverse geographies, time periods, and epidemiological contexts. A further potential limitation is that, without a gold standard diagnostic test, sensitivity and specificity estimates may be biased if the tests are less sensitive and/or specific for shared reasons. The hierarchical conditional dependence model we used accounted for this pairwise dependence and increased uncertainty around our estimates accordingly. This approach also allowed us to pool test performance estimates across studies from 4 countries. Thus, to our knowledge, we adjusted our estimates for test sensitivity and specificity using the best generic estimates available. Still, we likely overestimated sensitivity of culture for settings where samples had to be sent to a reference lab. Variation in the timing of tests in relation to when sample was taken could mean that one sensitivity and specificity estimate per diagnostic method is not appropriate. For example, a 2023 study in Haiti found that stool culture had a sensitivity of 33% during the waning phase of the 2018 to 2019 cholera outbreak [142], which is much lower than previous estimates. Overall, we have high confidence in our average estimates of *V. cholerae* positivity, despite the difficulty of accurately estimating positivity in a new location/time/setting without confirmation tests.

These findings have several implications for cholera surveillance policy. The GTFCC defines suspected cholera in areas where an outbreak has not yet been reported as acute watery diarrhea and severe dehydration or death in individuals 2 years and older [17]. Our finding that setting any minimum age increases specificity for identifying a true *V. cholerae* infection in suspected cases supports using an age restriction in this case definition. The February 2023

interim guidance from the GTFCC on cholera surveillance provides concrete recommendations for systematic and frequent testing of suspected cholera cases at the health facility or surveillance unit scale [17]. Our finding of high variability in positivity across settings and times lends support to these recommendations of systematically generating local data that can be used to scale suspected to true cholera. Our finding that high-quality sampling also increases specificity for *V. cholerae* suggests that systematically selecting cases to test is important for accurately evaluating endemic cholera. Finally, that *V. cholerae* positivity was lower during non-outbreak surveillance suggests that systematic confirmation testing is additionally important for understanding cholera burden and epidemiology in endemic, non-outbreak settings where cocirculation of other enteric pathogens is common.

These estimates of *V. cholerae* positivity address one part of the challenge in establishing the true burden of cholera: cases that are overcounted due to nonspecific suspected case definitions. A crucial next step will be to estimate missed cases due to care seeking and poor clinical surveillance. This could be done in part through systematically synthesizing data from studies of care seeking behavior for diarrheal symptoms (e.g., [143,144]), including where potential cholera cases seek care (e.g., at pharmacies, traditional healers, or hospitals). This could additionally be done through population representative surveys and active case finding, similar to studies conducted in Haiti [145] and Tanzania [146], respectively, which demonstrated higher mortality rates associated with cholera than had been reported through passive surveillance. Together, these studies will help to understand whether and to what degree missed cholera cases compensate for the biases described here in overcounting.

Ultimately, a better understanding of *V. cholerae* positivity will help us move toward estimates of true cholera incidence and mortality. Given the large heterogeneity between studies, it will be important to do this in a way that accounts for variation in *V. cholerae* positivity between sites. Moreover, the proportion of suspected cholera cases missed because of milder symptoms or barriers to healthcare seeking needs to be estimated and accounted for. Such estimates will provide crucial information to guide the allocation of limited resources such as vaccines in a way that most effectively supports cholera prevention and control.

## Supporting information

**S1 Checklist. Preferred Reporting Items for Systematic Reviews and Meta-Analyses (PRISMA) 2020 Checklist.**
(PDF)

**S1 Appendix. Supporting information.** Detailed methods, including systematic review search terms and the full statistical model, as well as additional figures and tables.
(PDF)

**S1 Data. Full dataset.** Excel sheet with the complete data extracted from all 131 studies that met the inclusion criteria (tab 1) as well as all variable descriptions (tab 2). Data extracted from the 119 nonoverlapping studies included in the main analysis dataset can be found by filtering for the values "1" in the column "Primary dataset."
(XLSX)

## Acknowledgments

We thank Morgane Dominguez for feedback on this manuscript, Lori Rosman for assistance developing the literature search strategy, and Javier Perez-Saez for feedback on the analytical methods.

## Author Contributions

**Conceptualization:** Kirsten E. Wiens, Andrew S. Azman.

**Data curation:** Kirsten E. Wiens, Hanmeng Xu, Kaiyue Zou, John Mwaba, Maya N. Demby, Elizabeth C. Lee, Andrew S. Azman.

**Formal analysis:** Kirsten E. Wiens, Andrew S. Azman.

**Funding acquisition:** Andrew S. Azman.

**Investigation:** Kirsten E. Wiens, Hanmeng Xu, Andrew S. Azman.

**Methodology:** Kirsten E. Wiens, Hanmeng Xu, Justin Lessler, Elizabeth C. Lee, Andrew S. Azman.

**Project administration:** Kirsten E. Wiens, Hanmeng Xu, Andrew S. Azman.

**Resources:** Andrew S. Azman.

**Supervision:** Andrew S. Azman.

**Validation:** Kirsten E. Wiens, Hanmeng Xu.

**Visualization:** Kirsten E. Wiens.

**Writing – original draft:** Kirsten E. Wiens, Hanmeng Xu, Andrew S. Azman.

**Writing – review & editing:** Kirsten E. Wiens, Hanmeng Xu, Kaiyue Zou, John Mwaba, Justin Lessler, Espoir Bwenge Malembaka, Maya N. Demby, Godfrey Bwire, Firdausi Qadri, Elizabeth C. Lee, Andrew S. Azman.

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
