## [Editor Report · Decision Letter 0]

27 Oct 2022

Dear Dr Azman, 

Thank you for submitting your manuscript entitled "Towards estimating true cholera burden: a systematic review and meta-analysis of Vibrio cholerae positivity" for consideration by PLOS Medicine.

Your manuscript has now been evaluated by the PLOS Medicine editorial staff as well as by an academic editor with relevant expertise and I am writing to let you know that we would like to send your submission out for external peer review.

Please re-submit your manuscript within two working days, i.e. by Oct 31 2022 11:59PM.

Kind regards,

Philippa Dodd, MBBS MRCP PhD

PLOS Medicine

---

## [Decision Letter · Decision Letter 1]

17 Apr 2023

Dear Dr. Azman,

Thank you very much for submitting your manuscript "Towards estimating true cholera burden: a systematic review and meta-analysis of Vibrio cholerae positivity" (PMEDICINE-D-22-03502R1) for consideration at PLOS Medicine. 

[LINK]

In light of these reviews, I am afraid that we will not be able to accept the manuscript for publication in the journal in its current form, but we would like to consider a revised version that addresses the reviewers' and editors' comments. Obviously we cannot make any decision about publication until we have seen the revised manuscript and your response, and we plan to seek re-review by one or more of the reviewers. 

We expect to receive your revised manuscript by May 08 2023 11:59PM. Please email us (plosmedicine@plos.org) if you have any questions or concerns.

We look forward to receiving your revised manuscript. 

Sincerely,

Philippa Dodd, MBBS MRCP PhD

PLOS Medicine

plosmedicine.org

GENERAL

Please respond to all editor and reviewer comments detailed below in full.

* Please report your SR/MA according to the PRISMA guidelines provided at the EQUATOR site.

http://www.equator-network.org/reporting-guidelines/prisma/

 - Please provide the completed PRISMA checklist. When completing the checklist, please use section and paragraph numbers, rather than page or line numbers, as these often change in the event of publication.

 - Please add the following statement, or similar, to the Methods: "This study is reported as per the Preferred Reporting Items for Systematic Reviews and Meta-Analyses (PRISMA) guideline (S1 Checklist)."

** Was the study registered with PROSPERO? Please provide details.

*** Please update your search to the present time – PLOS Medicine requires that all systematic reviews are updated 

to within 6 months of an anticipated publication date

COMMENTS FROM THE ACADEMIC EDITOR

Congrats on collating and synthesising these important data. Would your team be able to elaborate further on the utility of using suspect cholera case definitions in outbreak and non-outbreak settings? It seems the these case definitions have performed well during outbreaks (with some recommendations made in lines 258-270) but that laboratory confirmation may be needed for sporadic cases?

COMMENTS FROM THE EDITORS

ABSTRACT

Please report your abstract according to PRISMA for abstracts, following the PLOS Medicine abstract structure (Background, Methods and Findings, Conclusions) http://www.plosmedicine.org/article/info:doi/10.1371/journal.pmed.1001419

Please provide the specific start and end dates of your search, data sources, types of study designs included, eligibility criteria, and synthesis/appraisal methods. 

Please ensure that all numbers presented in the abstract are present and identical to numbers presented in the main manuscript text.

Please quantify the main results p values as well as with 95% CIs. Please report as p <0.001 or where higher as p=0.002, for example. Suggest separating upper and lower bounds of 95% CIs with commas instead of hyphens as these can be confused with negative values. For example, lines 56-57, suggest “…1.64 (95% Credible Interval: 1.06,2.52; p< or =)

Please include numerators and denominators used to derive percentages

Line 56 – do you present odds ratios here? Please clarify/define numerical values for the reader

Please include any important variables that are adjusted for in the analyses.

In the last sentence of the Abstract Methods and Findings section, please describe the main limitation(s) of the study's methodology.

Abstract Conclusions:

Please address the study implications without overreaching what can be concluded from the data; the phrase "In this study, we observed ..." may be useful.

Please interpret the study based on the results presented in the abstract, emphasizing what is new without overstating your conclusions.

Please avoid vague statements such as "these results have major implications for policy/clinical care". Mention only specific implications substantiated by the results.

AUTHOR SUMMARY

At this stage, we ask that you include a short, non-technical Author Summary of your research to make findings accessible to a wide audience that includes both scientists and non-scientists. The author summary should consist of 2-3 succinct bullet points under each of the following headings:

• Why Was This Study Done? Authors should reflect on what was known about the topic before the research was published and why the research was needed.

• What Did the Researchers Do and Find? Authors should briefly describe the study design that was used and the study’s major findings. Do include the headline numbers from the study, such as the sample size and key findings. 

• What Do These Findings Mean? Authors should reflect on the new knowledge generated by the research and the implications for practice, research, policy, or public health. Authors should also consider how the interpretation of the study’s findings may be affected by the study limitations.

The Author Summary should immediately follow the Abstract in your revised manuscript. This text is subject to editorial change and should be distinct from the scientific abstract. Please see our author guidelines for more information: https://journals.plos.org/plosmedicine/s/revising-your-manuscript#loc-author-summary

INTRODUCTION

The editorial team agree that it would be helpful to justify/clarify how measuring the ‘burden’ of cholera is helpful considering it occurs most often in the context of sporadic outbreaks (peaks & troughs).

Please indicate whether your study is novel and how you determined that. If there has been a previous systematic review of the evidence, please refer to and reference that review and indicate whether it supports the need for your study.

METHODS and RESULTS

Please justify why your search start date was 2000

As above please ensure you update the search to the present time.

PLOS Medicine encourages inclusion of all non-English language studies. Please justify why studies reported in languages other than “...English, French, Spanish, and Chinese…” were excluded (line 117). 

Studies reported on Medrix are not peer reviewed publications. It is not clear how many (if any) were included in the analyses. We suggest that these are excluded from the main analyses.

Please be reminded to update the manuscript accordingly including the PRISMA flowchart, figures, tables as necessary

As above, please include p values where you report 95% CIs. Please provide the statistical tests used to determine p values. Please report as p <0.001 or where higher as p=0.002, for example. 

Please ensure consistency when reporting upper and lower 95% CI bounds in the abstract hyhens are used, here the word “to” when negative values are reported. We suggest the use of commas throughout.

When referring to ‘odds’ do you mean odds ration (OR)? Please clarify and define numerical values as necessary for the reader.

FIGURES

To make your figures more accessible to those with colour blindness, please consider avoiding the use of green and/or red.

Figure 1 – details here are inconsistent with those in the abstract and main manuscript text, please revise in line with the above comments.

Figure 2 - Please clearly indicate in the figure caption the meaning of the boxes and whiskers

Figure 3 – Please clearly indicate in the figure caption the meaning of the dots and lines 

TABLES

Table 1 – please define PCR, RDT, please define the total number of studies as n= in the column header. Table 1 caption – in general this is a bit confusing “study-country-periods” is mentioned more than once but not in the table, what does it mean? Please revise/clarify for the reader such that the table contents are clearly defined without the need to refer to the manuscript text

There is 1 study defined as ‘other’ considering it is a single study would it be helpful to simply define it?

DISCUSSION

Please present and organize the Discussion as follows: a short, clear summary of the article's findings; what the study adds to existing research and where and why the results may differ from previous research; strengths and limitations of the study; implications and next steps for research, clinical practice, and/or public policy; one-paragraph conclusion. Please avoid the use of sub-headings such that discussion reads as a single continuous piece of prose.

REFERENCES

In the bibliography, please ensure that up to, but no more than, 6 author names are listed followed by et al, in the event that more that 6 authors contribute to an individual study. Journal name abbreviations should be those found in the National Center for Biotechnology Information (NCBI) databases. 

Please see our website for other reference guidelines https://journals.plos.org/plosmedicine/s/submission-guidelines#loc-references

SUPPLEMENTARY FILES

SUPP. METHODS

please ensure consistency with main manuscript text and figures

SUPP. FIGURES

Figure 1 & 2 – Please confirm that the appropriate usage rights apply to the use of this map. Please see our guidelines for map images: https://journals.plos.org/plosmedicine/s/figures#loc-maps

SUPP. TABLES

Please ensure all abbreviations are defined in the captions or an appropriate footnote (PCR/RDT, for example)

Please consider the use of commas to separate upper and lower bounds as hyphens can be confused with reporting of negative values.

Comments from the reviewers:

Reviewer #1: Thanks for the opportunity to review your manuscript. My role is as a statistical reviewer, so my review concentrates on the study design, data, and analysis that are presented. I have put general questions first, followed by queries relevant to a specific section of the manuscript (with a page/line reference).

This study is a systematic review and meta-analysis that examines test positivity for cholera. The systematic review is broad and includes a large pool of studies that met quality requirements. A Bayesian approach to test positivity is used, and includes an assessment of factors (surveillance, test type etc.) that may be associated with test positivity. This is out of my own usual working area, but I was convinced that an understanding of what affects test positivity rates is necessary background to make improvements to the quality of cholera testing. There were many overlapping studies (where the same patients contribute the same data to different studies), and this was dealt with by selecting the more desirable study (in terms of quality and size). While not perfect, this seems to be a reasonable way of dealing with the overlap. The analysis is Bayesian latent class model that takes into account the correlation of measures of test performance within the same study. The manuscript is clearly written, and I found the supplementary materials were excellent in helping me to understand the analysis (nice figures e.g. S4). 

The methods sections mentions has JAGs being used for analysis, but in the supplementary methods Stan is also mentioned. To clarify, was JAGS used for the latent class part of the analysis and Stan for the hierarchical meta-analysis? This is what it looks like from the github site. 

Are there terms to describe the types of cholera studies that met the inclusion criteria? E.g 'high quality' or 'population-based'? 

The analysis uses beta(1,1,) priors, i.e. 'flat prior', where all values of prevalence, sensitivity, and specificity are equally likely with varying upper and lower bounds. I would argue that this is probably not strictly 'uninformative' and it might be better to describe them differently, e.g. flat prior. I can follow the truncation bounds for some of the priors, but not others. For example, I wasn't clear why the lower bound of truncation for specificity was different between PCR and RDT. Some explanation of why these truncation points were chose for each parameter would be helpful.

R-hat is checked for model convergence, where any other diagnostic checks from analysis (e.g. trace plots, divergence plots) performed? 

Table S1 - should the last row 'Specificity of PCR uspec(RD)' actually be 'Specificity of RDT'?

Figure 2. With the number of studies contributing data I found it hard to get an understanding about differences in distribution of positivity according to the stratifying variables and quality of sampling. I wonder if something similar but with probability density plots (or violin or ridgeline plots) instead of box plots might allow more information to come out of this visualisation. 

Reviewer #2: 

Dear 

General remark:

The study covers a relevant topic and is well written in most parts. However, the manuscript has some major limitations. This systematic review and meta-analysis does not fulfil the requirements of the Preferred Reporting Items for Systematic Reviews and Meta-Analyses guidelines that may help to improve the manuscript. Please add a prisma cheklist as supplemantary file. 

Reviewer #3: Thank you for considering me as a reviewer for this very interesting and timely analysis. Given the surge of cholera globally in the recent years, the study is contextual and adds to the evidence base. The key finding, of significantly higher odds of detecting cholera during outbreak-linked surveillance is intuitively expected, but pegging an estimated value will possibly help in understanding reported cases better (especially in outbreak-prone areas). 

The discussion section lays out the shortcomings in the analysis, but it would be interesting if the authors could reflect on the policy implications of their findings, specifically, how can these findings be used by program managers and policymakers in cholera endemic and outbreak prone countries. This is especially important considering the rapid surge and spread of cholera in countries which have not seen a significant burden of cholera cases for several years (some even for decades). 

The missed burden of cholera is significant, as in all the settings where cholera is a public health problem, healthcare services are often accessed through informal routes. Without an estimate of these cases, asymptomatic or mild-to-moderately symptomatic cases and missed cases, a more holistic understanding of the true burden will not be forthcoming. The authors recommend that these should be accounted for using local contexts/data. Would it be possible for the authors to provide a little bit more framing around this, so that policy implementers in countries with novel and extended cholera outbreaks can try to identify a closer, locally valid estimate?

[LINK]

---

## [Decision Letter · Decision Letter 2]

12 Aug 2023

Dear Dr. Azman,

Thank you very much for re-submitting your manuscript "Towards estimating true cholera burden: a systematic review and meta-analysis of Vibrio cholerae positivity" (PMEDICINE-D-22-03502R2) for consideration at PLOS Medicine.

I have discussed the paper with our academic editor and it was also seen again by one reviewer. I am pleased to tell you that, provided the remaining editorial and production issues are fully dealt with, we expect to be able to accept the paper for publication in the journal.

[LINK]

Please let me know f you have any questions, and we look forward to receiving the revised manuscript.   

Sincerely,

Richard Turner PhD, for Philippa Dodd, MBBS MRCP PhD

Consulting Editor, PLOS Medicine

plosmedicine@plos.org

Requests from Editors:

GENERAL

Thank you for your detailed responses to previous editor and reviewer comments. Please see below for further comments that we require you address in full prior to publication.

COMPETING INTERESTS

All authors must declare their relevant competing interests per the PLOS policy, which can be seen here:

https://journals.plos.org/plosmedicine/s/competing-interests

For authors with ties to industry, please indicate whether any of the interests has a financial stake in the results of the current study.

We ask you to revise the title to better match journal style and suggest: "Estimating the proportion of Vibrio cholerae infections among suspected cholera cases: A systematic review and meta-analysis”.

ABSTRACT

Please combine the methods and findings sections into one section, ‘Methods and findings’

Line 54 – ‘V. cholerae positivity was lower in studies with representative sampling and lower minimum ages in suspected case definitions.’ The use of the term lower (I think) in two different contexts is confusing, please revise for clarity.

Line 60 – please remove ‘and the resolution of the data’

Line 71 – please remove the funding statement and include only in the manuscript submission form.

In the main methods section (line 155) you state ‘…January 1, 2000 to reflect contemporary patterns in cholera positivity…’ please include the same or similar in the abstract to clearly justify your choice of search start date.

AUTHOR SUMMARY

Thank you for including an author summary, which reads very nicely. 

Line 91 – please place as the final bullet point of this sub-section.

INTRODUCTION

Line 112 - please define ‘PCR’ at first use for the reader.

METHODS and RESULTS

Line 202 – please move this statement to the beginning of the methods section.

Currently, the ethics approval appears to be quoted both at the start and end of the Methods section: please de-duplicate. 

TABLES

Please include a table which summaries the basic information of the studies utilized for your review – author, year of publication, country, study type/design, number of ppts, diagnostic test as a minimum.

Please include an assessment of study quality.

Table 1 – please provide an appropriate caption which clearly details the table content without the need to refer to the text. What do you mean by ‘Number of observations’ the ‘observations’ confuses me. Could it simply read number?

FIGURES

In the captions for example line 922 you refer to ‘observations’ here also. This term is not used in the text. Do you mean observed cases? Please clarify/amend and in consistency with table 1.

Figures 2, 3 and 4 please define all abbreviations in either a footnote or in the figure caption – PCR, RDT, IQR, V.cholera

Figure 3 – please detail the meaning of the dots and lines in the figure caption.

DISCUSSION

Lines 48 and 443 – please remove these statements from the end of the discussion and include only in the manuscript submission form they will be compiled as metadata.

SUPPORTING INFORMATION

Please cite, label and upload your Supporting Information as outlined here: https://journals.plos.org/plosmedicine/s/supporting-information

Please ensure that the reference format follows that of our guidance which can be found here. https://journals.plos.org/plosmedicine/s/submission-guidelines#loc-references

Figures – please define abbreviations RDT and PCR throughout in the caption or in a footnote.

Please use the form "PLoS" in the reference list; and revisit reference 88, which appears to be missing publication information.

SOCIAL MEDIA

To help us extend the reach of your research, please detail any Twitter handles you wish to be included when we tweet this paper (including your own, your coauthors’, your institution, funder, or lab) in the manuscript submission form when you re-submit the manuscript.

Comments from Reviewers:

*** Reviewer #1: 

Thanks for the revised manuscript and replies to my original review.

With the addition of the new data, some of the STAN models needed modification. This sounds like a reasonable strategy to deal with the heterogeneity. 

The manuscript looks good from my perspective, the figures are great (colour scheme looks fine to me).

***

[LINK]

---

## [Editor Report · Decision Letter 3]

25 Aug 2023

Dear Dr Azman, 

On behalf of my colleagues and the Academic Editor, Dr Amitabh Suthar, I am pleased to inform you that we have agreed to publish your manuscript "Estimating the proportion of clinically suspected cholera cases that are true Vibrio cholerae infections: A systematic review and meta-analysis" (PMEDICINE-D-22-03502R3) in PLOS Medicine.

Prior to publication we require that you make the following changes:

* Line 49 - should read ‘April 19, 2023’

* Line 250 - please remove the data availability statement from the main manuscript and include only in the manuscript submission form.

* Line 252 - sentence beginning ‘Extracted data…’ suggest moving this to an appropriate part of the results section. Line 263 perhaps?

PRESS

Sincerely, 

Philippa Dodd, MBBS MRCP PhD 

PLOS Medicine